# Fundamental limits of parasitoid-driven host population suppression: Implications for biological control

**Abhyudai Singh** *

Departments of Electrical and Computer Engineering, Biomedical Engineering and Mathematical Sciences, University of Delaware, Newark, DE, United States of America

* absingh@udel.edu

**Data Availability Statement:** All relevant data and methods are within the paper and its Supporting information files.

**Funding:** The authors received no specific funding for this work.

## Abstract

Parasitoid wasps are increasingly being used to control insect pest populations, where the pest is the host species parasitized by the wasp. Here we use the discrete-time formalism of the Nicholson-Bailey model to investigate a fundamental question—are there limits to parasitoid-driven suppression of the host population density while still ensuring a stable coexistence of both species? Our model formulation imposes an intrinsic self-limitation in the host's growth resulting in a carrying capacity in the absence of the parasitoid. Different versions of the model are considered with parasitism occurring at a developmental stage that is before, during, or after the growth-limiting stage. For example, the host's growth limitation may occur at its larval stage due to intraspecific competition, while the wasps attack either the host egg, larval or pupal stage. For slow-growing hosts, models with parasitism occurring at different life stages are identical in terms of their host suppression dynamics but have contrasting differences for fast-growing hosts. In the latter case, our analysis reveals that wasp parasitism occurring after host growth limitation yields the lowest pest population density conditioned on stable host-parasitoid coexistence. For ecologically relevant parameter regimes we estimate this host suppression to be roughly 10-20% of the parasitoid-free carrying capacity. We further expand the models to consider a fraction of hosts protected from parasitism (i.e., a host refuge). Our results show that for a given host reproduction rate there exists a critical value of protected host fraction beyond which, the system dynamics are stable even for high levels of parasitism that drive the host to arbitrary low population densities. In summary, our systematic analysis sheds key insights into the combined effects of density-dependence in host growth and parasitism refuge in stabilizing the host-parasitoid population dynamics with important implications for biological control.

## 1 Introduction

Parasitoids are often used as agents of biological control to manage insect pets in forest and agricultural ecosystems. The primary goal of such biological control is to suppress the pest population density through the introduction of natural enemies, such as parasitoids, and hence minimize or eliminate the usage of insecticides. This goal sets up an interesting tradeoff

**Competing interests:** The authors have declared that no competing interests exist.

on the level of parasitism—while a certain minimal level is needed for parasitoid establishment, high levels of parasitism can destabilize the system eventually leading to parasitoid extinction [1]. This contribution uses population dynamic models to rigorously quantify these tradeoffs and determine the optimal parasitism levels that yield the lowest host population density while still ensuring the stable coexistence of both species.

There is a long-standing tradition of modeling host-parasitoid population dynamics using discrete-time models [1–6]. This is primarily motivated by populations living in the temperate regions of the world where annual insect life stages are synchronized by season and reproduction occurs at specific times in the year. A typical life cycle is illustrated in Fig 1 where adult hosts in a given year emerge and lay eggs that hatch into larvae. Host larvae feed and grow on resources and then pupate to metamorphosize as adults the following year. At one of its life stages, the host is parasitized by a parasitoid wasp (for example, in Fig 1 the wasps attack host eggs). Adult female parasitoids locate and oviposit an egg into the host that hatches into a juvenile parasitoid. The juvenile develops using the host's body as a food source and goes on to transform into an adult parasitoid the next year, killing the parasitized host in the process [7, 8]. It is important to point out that the host is only needed for juvenile development, and adult parasitoids are free-living insects.

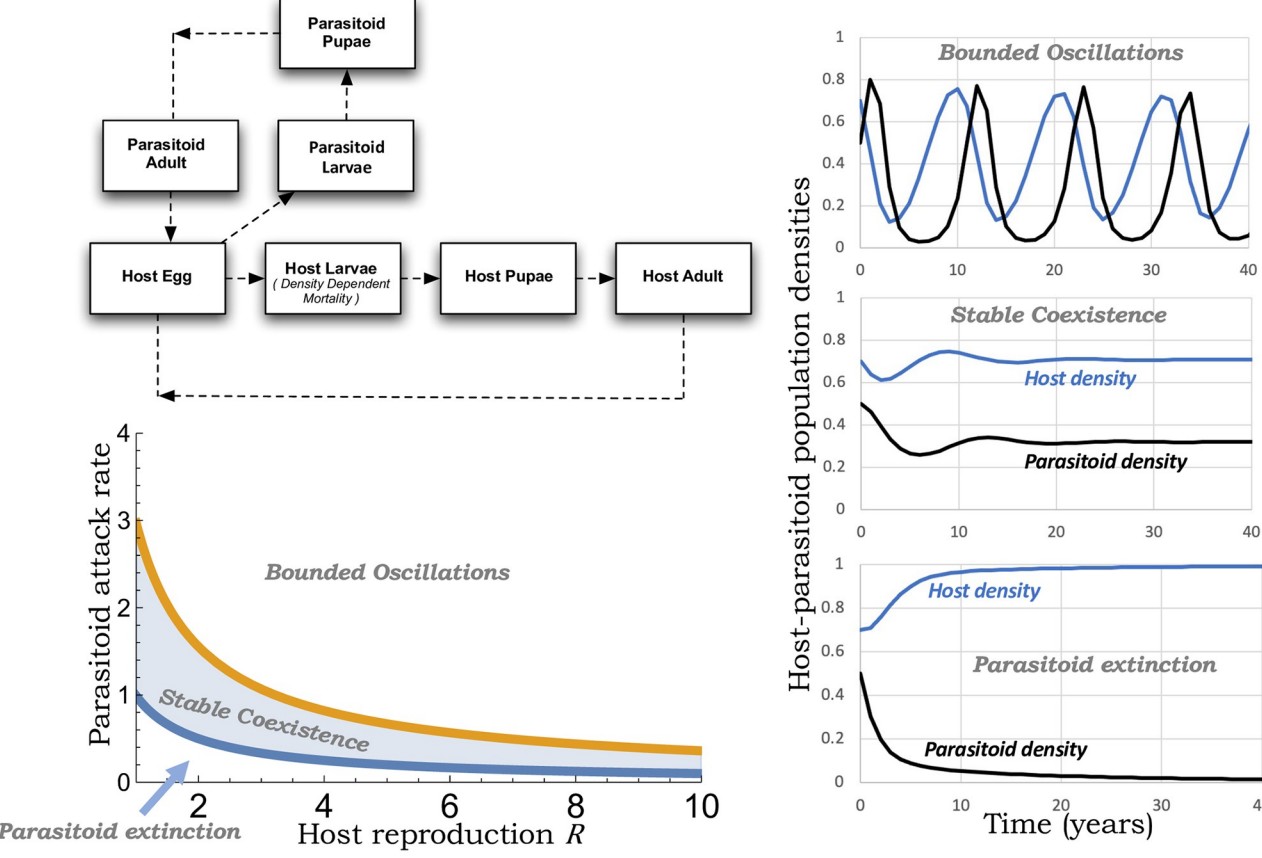

**Fig 1. Schematic of the insect life cycles, with parasitoids attacking the host egg stage.** Self-limitation in host growth due to resource competition is assumed to occur at the larval stage and is captured by a density-dependent mortality rate. *Bottom-left*: Dynamical behaviors of the model (18) for $k = K = 1$, where for a given host reproduction rate $R$, increasing levels of parasitism results in—parasitoid extinction (low attack rate), stable coexistence of both the host and its parasitoid (intermediate attack rate), and bounded oscillations in population densities (high attack rate). *Right*: Representative population trajectories for the different dynamical scenarios with parameters taken as $K = k = 1$, $R = 2$, and $c_p = 0.49, 0.8, 1.7$ in (18) from bottom to top.

A simple model capturing the ecological dynamics of this interaction is given by

$$H_{t+1} = RH_t f(P_t), \tag{1a}$$

$$P_{t+1} = kRH_t(1 - f(P_t)) \tag{1b}$$

where $H_t$ and $P_t$ are the adult host and parasitoid population densities in year $t$, respectively. If $R > 1$ is the number of eggs produced by each adult host, then $RH_t$ is the host density vulnerable to parasitism. In the text, $R$ is referred to as the *host reproduction rate*. The *escape response* $f(P_t)$ is the fraction of hosts escaping parasitism to become next year's adult hosts with $f(0) = 1$. Following this model, $RH_t[1 - f(P_t)]$ is the density of parasitized hosts with $k \geq 1$ adult female parasitoids developing per infected host. A classical form of (1) in the Nicholson-Bailey model [9]

$$f(P_t) = \exp(-c_p T P_t) := e^{-c_p T P_t} \tag{2}$$

where $T$ is the duration of the host vulnerable stage, and $c_p > 0$ is the *parasitoid attack rate*. One can interpret the attack rate as $1/c_p$ being the average time taken by a parasitoid to forage, locate, and parasitize a given host. Key assumptions underlying the Nicholson-Bailey model is that the parasitoids are search-limited (but not egg-limited) and have fast handling times.

The Nicholson-Bailey model has a single non-trivial fixed point

$$H^* = \frac{\log(R)}{(R-1)kc_p T}, \quad P^* = \frac{\log(R)}{c_p T}. \tag{3}$$

that is unstable resulting in diverging population density oscillations [9]. Here and throughout the manuscript log represents the natural logarithm. A Type II functional response can be incorporated in (3) by replacing $c_p$ with

$$\frac{c_p}{1 + c_p T_h RH_t} \tag{4}$$

where $T_h$ is the handling time, and this has been shown to further destabilize the population dynamics [1, 10, 11]. A multitude of mechanisms are known to stabilize population dynamics and they can be classified into two types [10, 12]:

1. The first class of mechanisms includes those stabilizing factors, where the *escape response* $f(P_t)$ only depends on the parasitoid density. One such example is a proportional host refuge, where a fraction $\mu$ of hosts are protected from parasitism due to elevated host defenses or being inaccessible to parasitoids. In this case, the Nicholson-Bailey escape response is modified to

$$f(P_t) = \mu + (1 - \mu)\exp(-c_p P_t) \tag{5}$$

[13]. Other stabilizing mechanisms that fall in this category of a parasitoid-dependent $f(P_t)$ include interference between parasitoids [14], host-to-host differences in susceptibility to parasitism [15–20], and aggregation of parasitoid attacks on high-risk hosts [21–23]. In all these cases, a *necessary and sufficient* condition for stable host-parasitoid coexistence is

$$\frac{dH^*}{dR} > 0, \tag{6}$$

i.e., the adult host equilibrium density (in the parasitoid's presence) is an increasing

function of $R$ [24]. This is also reflected in the instability of the Nicholson-Bailey fixed point (3) where $H^*$ is a decreasing function of $R$.

2. The second class of mechanisms includes a Type III functional response where the parasitoid attack rate increases (or accelerates) in response to higher host density [25, 26]. Such responses have been reported for several parasitoids [27, 28], and involve a change in behavior where the consumer is able to exploit the resource more efficiently at higher densities [29]. For example, parasitoids can have much faster handling times [30], or spend more time searching for hosts at higher host densities [31]. A key difference with the earlier case is that here the escape response depends on both the host and parasitoid densities and stability arises with the adult host density being a decreasing function of $R$ [32].

In this contribution, we primarily focus on the case of a constant attack rate $c_p$ (as in the Nicholson-Bailey model), and later in the manuscript we investigate the impact of proportional host refuge as outlined in the first class of mechanisms.

The Nicholson-Bailey model has no intrinsic self-limitation in host growth with geometric expansion in host numbers

$$H_{t+1} = RH_t \Rightarrow H_t \propto R^t, \tag{7}$$

in the parasitoid's absence. Previous works have shown the stabilizing effects of including density-dependent self-limitation in host growth [33–38]. Motivated by this, we model the parasitoid-free host's population dynamics as per the Beverton-Holt model

$$H_{t+1} = \frac{RH_t}{1 + c_h RH_t} \tag{8}$$

that has been previously reported in the context of intraspecific competitions [39–41]. The parameter $c_h > 0$ quantifies the strength of this competition and (8) has a stable fixed point

$$H^* = \frac{R-1}{c_h R} = K \Rightarrow c_h = \frac{R-1}{KR} \tag{9}$$

for all values of $R$, $c_h$ and $K$ denotes the carrying capacity. Without loss of any generality, we assume that this growth limitation acts at the host larval stage and consider different parasitism scenarios:

- Parasitoids attack host eggs—parasitism occurs at a stage before host growth limitation.

- Parasitoids attack host larvae. This leads to two different models depending on whether intraspecific competition acts only on the unparasitized larvae or on all (unparasitized and parasitized) larvae.

- Parasitoids attack host pupae—parasitism occurs at a stage after host growth limitation.

A key question driving this investigation is how much suppression in host density below its parasitoid-free carrying capacity is possible while still ensuring the stable coexistence of both species. Are there parameter regimes where the population dynamics of host suppression is invariant to the relative timing of parasitism with respect to host growth limitation? For regimes where this relative timing is critically important, which scenario provides the most efficient suppression of host density? Finally, we expand the study by coupling host growth limitation with an additional stabilizing factor of host refuge to investigate their combined effects on stable coexistence.

## 2 General model formulation and analysis

We start by reviewing a mechanistic derivation of the Beverton-Holt model using the semi-discrete approach, where update rules in discrete-time models are obtained by solving a system of continuous-time differential equations [42–49]. Let $L(t, \tau)$ denote the host larval density in year $t$ at time $\tau \in [0, 1]$ within the larval stage, where $\tau = 0$ and $\tau = 1$ correspond to the start and end of the stage, respectively. We consider a per capita density-dependent larval mortality rate $c_h L(t, \tau)$ that scales linearly with the population density and acts continuously throughout the stage. This mortality could be a result of intraspecific competition for resources or predation by natural enemies other than the parasitoid in consideration. Then, the larval density decays continuously as per the ordinary differential equation

$$\frac{dL(t, \tau)}{d\tau} = -c_h L(t, \tau)^2. \tag{10}$$

Solving (10) yields

$$L(t, \tau) = \frac{L(t, 0)}{1 + c_h \tau L(t, 0)}, \tag{11}$$

and using the initial condition $L(t, 0) = RH_t$ together with $H_{t+1} = L(t, 1)$ (i.e., surviving larvae at the end of the stage become next year's adults) results in the Beverton-Holt model (8).

In the parasitoid's presence, (8) transforms to

$$H_{t+1} = RH_t f(RH_t, P_t) \tag{12a}$$

$$P_{t+1} = kg(RH_t, P_t), \tag{12b}$$

where $f$ and $g$ are continuously differentiable functions that depend on the host and parasitoid population densities with

$$f(RH_t, 0) = \frac{1}{1 + c_h RH_t}, \quad g(RH_t, 0) = 0. \tag{13}$$

Apart from the trivial fixed point that excludes both species, the model's fixed points are given by simultaneously solving

$$f(RH^*, P^*) = \frac{1}{R}, \quad P^* = kg(RH^*, P^*), \tag{14}$$

where $H^*$ and $P^*$ represent the host and parasitoid equilibrium densities, respectively. One of these fixed points corresponds to parasitoid extinction ($P^* = 0$) and the host at its carrying capacity ($H^* = K$). We are primarily interested in the existence of alternative fixed points that allow for the stable coexistence of both the host and the parasitoid—and directly connected to it—what is the lowest possible value of $H^*/K$ that quantifies the limit of parasitoid-mediated suppression of pest population density. We next present stability analysis tools for nonlinear systems of the form (12).

The stability of the fixed point can be assessed by linearizing the nonlinearities in (12) for small perturbations around the fixed point. This process yields the linear discrete-time

dynamical system

$$h_t = H_t - H^*, \quad p_t = P_t - P^* \tag{15a}$$

$$\begin{bmatrix} h_{t+1} \\ p_{t+1} \end{bmatrix} = A \begin{bmatrix} h_t \\ p_t \end{bmatrix}, \quad A = \begin{bmatrix} a_{00} & a_{01} \\ a_{10} & a_{11} \end{bmatrix} \tag{15b}$$

$$a_{00} = 1 + RH^* \frac{\partial f(RH_t, P_t)}{\partial H_t}\big|_{H_t=H^*, P_t=P^*} \tag{15c}$$

$$a_{01} = RH^* \frac{\partial f(RH_t, P_t)}{\partial P_t}\big|_{H_t=H^*, P_t=P^*} \tag{15d}$$

$$a_{10} = k \frac{\partial g(RH_t, P_t)}{\partial H_t}\big|_{H_t=H^*, P_t=P^*} \tag{15e}$$

$$a_{11} = k \frac{\partial g(RH_t, P_t)}{\partial P_t}\big|_{H_t=H^*, P_t=P^*}. \tag{15f}$$

The fixed point is stable, if and only if, all the following three conditions hold

$$1 - tr(A) + det(A) > 0, \quad 1 + tr(A) + det(A) > 0, \quad 1 - det(A) > 0 \tag{16}$$

[50], where

$$tr(A) = a_{00} + a_{11}, \quad det(A) = a_{00}a_{11} - a_{01}a_{10}, \tag{17}$$

are the trace and determinant of the $2 \times 2$ Jacobian matrix $A$, respectively.

## 3 Parasitoids attack the host egg stage

We start with the scenario where the parasitoids parasitize the host eggs. As per the Nicholson-Bailey model, the unparasitized and parasitized host densities at the end of the egg stage are $RH_t \exp(-c_p P_t)$ and $RH_t(1 - \exp(-c_p P_t))$, respectively. While the parasitized eggs become next year's adult parasitoids, the unparasitized eggs hatch into larvae to face intraspecific competition and later develop into next year's adult hosts. This results in the following discrete-time model describing the host-parasitoid population dynamics

$$H_{t+1} = \frac{RH_t \exp(-c_p P_t)}{1 + c_h RH_t \exp(-c_p P_t)} \tag{18a}$$

$$P_{t+1} = kRH_t[1 - \exp(-c_p P_t)], \tag{18b}$$

where (18a) corresponds to (11) with initial condition $L(t, 0) = RH_t \exp(-c_p P_t)$. Here and in other models, for the sake of convenience, we assume the duration of the host vulnerable stage $T = 1$.

Standard stability analysis shows that the no-parasitoid fixed point ($H^* = K$, $P^* = 0$) is stable for

$$c_p < \frac{1}{kRK}. \tag{19}$$

One can also see from (18b) that a sufficient large attack rate $c_p > \frac{1}{kRK}$ is needed for parasitoid establishment ensuring population number increase from low densities (i.e., $P_{t+1}/P_t > 1$ when $H_t = K$ and $P_t \to 0$). When $c_p > \frac{1}{kRK}$, there exists a unique fixed point corresponding to the coexistence of both species that is given as the solution to

$$P^* = \frac{RKk(1 - \exp(-c_p P^*))(R - \exp(c_p P^*))}{R - 1} \tag{20a}$$

$$H^* = \frac{K(R - \exp(c_p P^*))}{R - 1}. \tag{20b}$$

Our analysis reveals that this fixed point is stable for

$$\frac{1}{kRK} < c_p < c_p^* \tag{21}$$

with attack rates above a critical value $c_p^*$ destabilizing the population dynamics. This critical value is obtained by solving

$$c_p^* k \exp(c_p^* P^*) K(R - \exp(c_p^* P^*)) = R - 1 \tag{22}$$

where $P^*$ in (22) is given by (20a) with $c_p = c_p^*$. Beyond $c_p > c_p^*$ the system exhibits bounded oscillations in population densities whose amplitude and time period increases with increasing attack rate. The range (21) of attack rates allowing for stable coexistence is illustrated in Fig 1 along with the representative population trajectories corresponding to different dynamical outcomes:

- Parasitoid extinction and the host at its carrying capacity for $c_p < \frac{1}{kRK}$.

- Stable host-parasitoid coexistence for $\frac{1}{kRK} < c_p < c_p^*$.

- Bounded oscillations in population densities for $c_p^* < c_p$.

The plot in Fig 1 showing the stability region was generated in Wolfram Mathematica, where for a given parameter set, $c_p^*$ was obtained by numerically solving (22) (see S1 File that has the Wolfram Mathematica code used for generating the stability region). The population density trajectories were plotted in Microsoft Excel by iteratively solving the discrete-time model (18). The lowest possible stable suppression of host density occurs when $c_p = c_p^*$ and is given by

$$\frac{H^*}{K} = \frac{R - \exp(z^*)}{R - 1} \tag{23}$$

where $z^*$ is the unique solution to

$$\exp(-2z^*)(\exp(z^*) - 1)R = z^*. \tag{24}$$

It is interesting to note that this limit of biological control only depends on $R$, and in this case, it increases with $R$ varying from 40%($R = 2$) to 55%($R = 10$). We next contrast this result with parasitism occurring at the host larval or pupal stage.

## 4 Parasitoids attack the host larval stage

When parasitoids parasitize the host larvae, both parasitism and density-dependent mortality from intraspecific competition occurs concurrently and continuously throughout the stage. Our previous work analyzed this case assuming that intraspecific competition only acts on the unparasitized larvae [42]. We review these results and also consider the scenario where both unparasitized/parasitized larvae compete for resources.

### 4.1 Density-dependent mortality on unparasitized larvae

Using the semi-discrete formalism for the mechanistic derivation of discrete-time models, the continuous changes in population densities are described by the ordinary differential equations

$$\frac{dL(t,\tau)}{d\tau} = -c_p L(t,\tau)P_t - c_h L(t,\tau)^2 \tag{25a}$$

$$\frac{dI(t,\tau)}{d\tau} = c_p L(t,\tau)P_t \tag{25b}$$

where $L(t,\tau)$ and $I(t,\tau)$ are the unparasitized and parasitized larval densities in year $t$ at time $\tau \in [0,1]$ within the stage. Solving (25) with $L(t,0) = RH_t$, $I(t,0) = RH_t$ yields the model

$$H_{t+1} = L(t,1) = \frac{RH_t \exp\left(-c_p P_t\right)}{1 + c_h RH_t \frac{1 - \exp\left(-c_p P_t\right)}{c_p P_t}} \tag{26a}$$

$$P_{t+1} = kI(t,1) = \frac{kc_p P_t}{c_h} \log\left[1 + c_h RH_t \frac{1 - \exp\left(-c_p P_t\right)}{c_p P_t}\right]. \tag{26b}$$

Here, a minimum attack rate

$$c_p > \frac{c_h}{k \log R}, \quad c_h = \frac{R-1}{KR}, \tag{27}$$

is needed for parasitoid establishment. Given this sufficiently large attack rate, there exists a unique fixed point

$$H^* = \left(\frac{\exp\left(\frac{c_h}{kc_p}\right) - 1}{R - \exp\left(\frac{c_h}{kc_p}\right)}\right) \frac{c_p P^*}{c_h}, \quad P^* = \frac{\log R - \frac{c_h}{kc_p}}{c_p}, \tag{28}$$

where both species are present. This coexistence equilibrium is stable for attack rates in the range

$$\frac{c_h}{k \log R} < c_p < \frac{c_h}{k\bar{z}}, \tag{29}$$

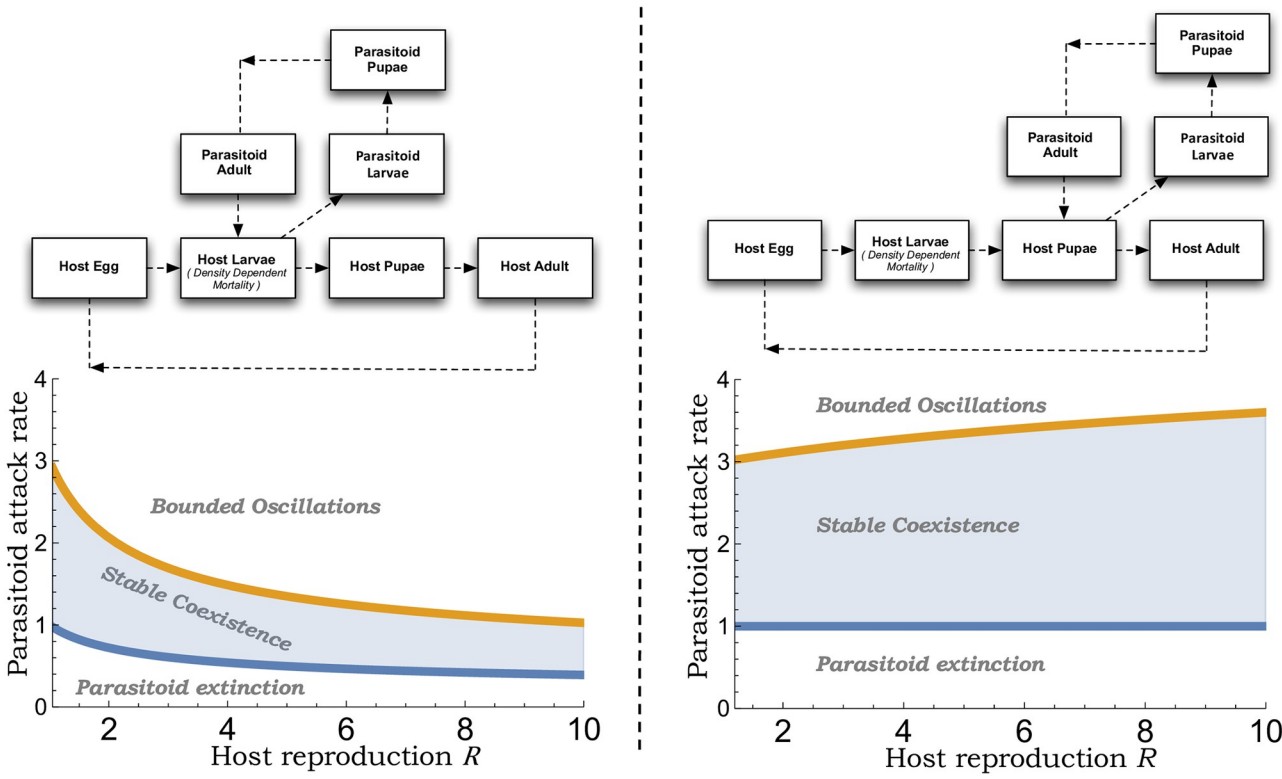

**Fig 2. Difference in population dynamics for larval vs. pupal parasitoids.** The top plots show the life cycles of a larval (left) and pupal (right) parasitoid with host growth limitation occurring at the larval stage. The bottom plots show the corresponding range of stabilizing attack rates given by (29) and (35) as a function of the host reproduction rate $R$. While the region of stability contracts with increasing $R$ for a larval parasitoid (left), it expands for a pupal parasitoid (right).

where $\bar{z}$ is the solution to

$$\bar{z} + 1 = \frac{R(\log R - \bar{z})}{R - \exp(\bar{z})}.$$

[42]. This range (29) is shown in Fig 2, with lower and higher values of $c_p$ resulting in parasitoid extinction and bounded oscillation, respectively.

The largest attack rate $c_p = \frac{c_h}{k\bar{z}}$ allowing a stable coexistence yields the lowest host density

$$\frac{H^*}{K} = \frac{(\bar{z} + 1)(\exp(\bar{z}) - 1)}{R - 1} \tag{30}$$

corresponding to a larval parasitoid. In contrast to egg parasitoids, this limit (30) slightly increases close to $R \approx 1$ (Fig 3), and then decreases with $R$ varying from 35% ($R = 2$) to 30% ($R = 10$).

## 4.2 Density-dependent mortality on all larvae

When both unparasitized and parasitized larvae have density-dependent mortality then (25) is altered to

$$\frac{dL(t,\tau)}{d\tau} = -c_p L(t,\tau) P_t - c_h L(t,\tau)(L(t,\tau) + I(t,\tau)) \tag{31a}$$

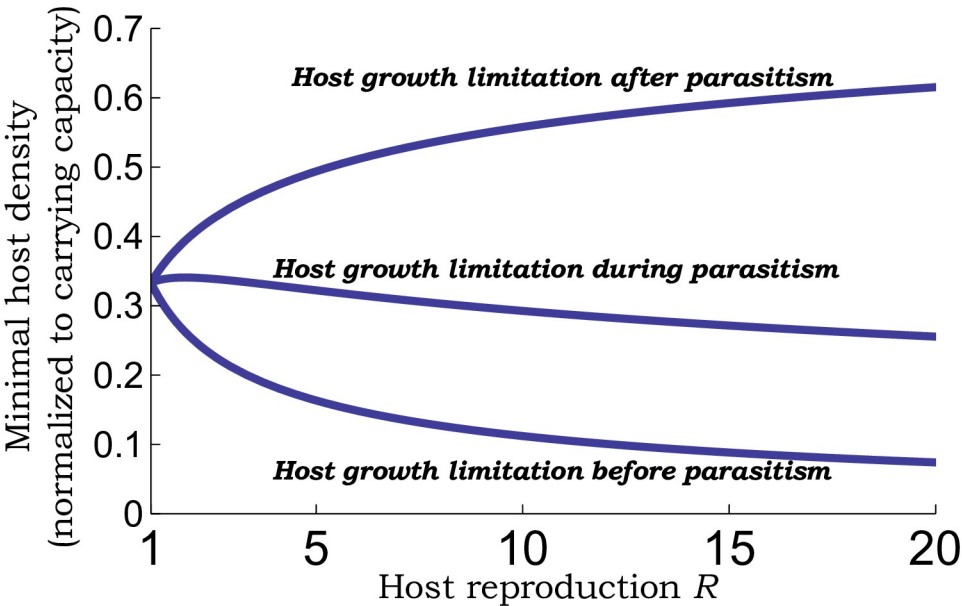

**Fig 3. Minimal possible host population density conditioned on stable host-parasitoid coexistence.** Plots of $H^*/K$ (host density with parasitoid normalized to its parasitoid-free carrying capacity) as a function of the host reproduction rate $R$ as given by (23) for an egg parasitoid (host growth limitation after parasitism), (30) for a larval parasitoid (host growth limitation occurring concurrently with parasitism), and (37) for a pupal parasitoid (host growth limitation before parasitism). All cases consider Beverton-Holt type host population dynamics (8) in the absence of the parasitoid with density-dependent self-limitation in growth assumed to occur at the host larval stage.

$$\frac{dI(t,\tau)}{d\tau} = c_p L(t,\tau)P_t - c_h I(t,\tau)(L(t,\tau) + I(t,\tau)), \tag{31b}$$

yielding the model

$$H_{t+1} = \frac{RH_t \exp(-c_p P_t)}{1 + c_h RH_t} \tag{32a}$$

$$P_{t+1} = \frac{kRH_t(1 - \exp(-c_p P_t))}{1 + c_h RH_t}. \tag{32b}$$

Incidentally, this is the exact same model as that for a pupal parasitoid. To see this, note that after intraspecific competition $RH_t/(1 + c_h RH_t)$ is the host density at the start of the pupal stage. While fraction $\exp(-c_p P_t)$ of host pupae escape parasitism to metamorphose into adult hosts, the other fraction $1 - \exp(-c_p P_t)$ are parasitized.

## 5 Parasitoids attack the host pupal stage

Population dynamics of a pupal parasitoid as described by the model (32) shares qualitatively similar dynamical outcomes to its counterparts (18) and (26), but with contrasting quantitative parameter ranges. The establishment of a pupal parasitoid requires a minimal attack rate

$$c_p > \frac{1}{kK} \tag{33}$$

that ensures a unique equilibrium

$$P^* = \frac{Kk(1 - \exp(-c_p P^*))(R - \exp(c_p P^*))}{R - 1} \tag{34a}$$

$$H^* = \frac{K \exp(-c_p P^*)(R - \exp(c_p P^*))}{R - 1} \tag{34b}$$

where both species are present. This fixed point is stable for attack rates in the range

$$\frac{1}{kK} < c_p < c_p^* \tag{35}$$

where $c_p^*$ is given by

$$c_p^* k \exp(c_p^* P^*) K(R - \exp(c_p^* P^*)) = (R - 1)R. \tag{36}$$

Plotting (35) as a function of $R$ in Fig 3 reveals key differences with previous cases:

- The range of stabilizing attack rates is much broader for a pupal parasitoid (Fig 2; right) as compared to an egg (Fig 1) or larval parasitoid (Fig 2; left).

- The range (35) expands with $R$—a fixed lower bound and an upper bound $c_p^*$ that increases with increasing $R$ (Fig 2; right).

- This expanding stability region is in sharp contrast to when parasitism occurs in earlier life stages, where with increasing $R$ both the lower/upper bound of stabilizing attack rate decreases with a contracting region of stability (Figs 1 & 2).

The lowest host population density corresponding to $c_p = c_p^*$ is given by

$$\frac{H^*}{K} = \frac{\exp(-z^*)(R - \exp(z^*))}{R - 1} \tag{37}$$

where $z^*$ is the unique solution to (24), and this limit is found to sharply decrease from 20% ($R = 2$) to 10%($R = 10$) (Fig 3).

## 6 Inclusion of host refuge

As mentioned in the Introduction, diverse ecological mechanisms are known to stabilize the Nicholson-Bailey model. Here we consider one such mechanism, where a fixed fraction $0 \le \mu < 1$ of hosts are protected from parasitism [13]. For example, these hosts could be in specific locations that are less accessible to parasitoids or have an elevated immune response against parasitism [51–54]. For example, the host *Bactrocera dorsalis* (oriental fruit fly) is parasitized by the pupal parasitoid *Dirhinus giffardii*. *B. dorsalis* larvae pupate below ground, and data shows that pupation depth determines the risk of parasitism—pupae further underground experience much lower rates of parasitism compared to pupae closer to the surface [55]. We focus on how such refuges work concertedly with the host's growth limitation to stabilize the host-parasitoid population dynamics.

## 6.1 Without host self-limitation

We first consider a host refuge fraction $\mu$ in the Nicholson-Bailey model resulting in

$$H_{t+1} = RH_t f(P_t), \tag{38a}$$

$$P_{t+1} = kRH_t(1 - f(P_t) \tag{38b}$$

where

$$f(P_t) = \mu + (1 - \mu) \exp(-c_p P_t) \tag{39}$$

is the fraction of hosts escaping parasitism [13]. Note that here $\mu > 1/R$ would result in unbounded population growth, and this is prevented in the next section by including a carrying capacity. When $\mu < 1/R$, (38) has a non-trivial fixed point given by

$$H^* = \frac{\log\left(\frac{(1-\mu)R}{(1-\mu R)}\right)}{(R-1)kc_p}, \quad P^* = \frac{\log\left(\frac{(1-\mu)R}{(1-\mu R)}\right)}{c_p}. \tag{40}$$

Since in (39) the fraction of host escaping parasitism only depends on $P_t$, (40) is stable iff (6) holds. Substituting $H^*$ from (40) in (6) it can be seen that there exists a critical refuge $\mu^*$ that *only* depends on $R$ and is given as the unique solution to

$$\frac{R-1}{R(1-\mu^* R)} = \log\left(\frac{R(1-\mu^*)}{1-\mu^* R}\right), \tag{41}$$

such that the fixed point (40) is stable when

$$\mu^* < \mu < 1/R, \tag{42}$$

for all values of $c_p$ and $k$. While $\mu^*$ cannot be analytically solved from (41), when $R \approx 1$, then using

$$\log z \approx \frac{2(z-1)}{z+1}, \quad z \approx 1 \tag{43}$$

we can approximate

$$\log\left(\frac{R(1-\mu^*)}{1-\mu^* R}\right) \approx \frac{2(R-1)}{R+1-2\mu^* R}, \quad R \approx 1, \tag{44}$$

in (41) to solve for $\mu^*$ yielding

$$\mu^* \approx \frac{1}{2R}, \quad R \approx 1. \tag{45}$$

The actual value of $\mu^*$ as obtained by numerically solving (41) is shown in Fig 4 (bottom orange line in the top-left plot). While as predicted by (45), $\mu^* \to 0.5$ as $R \to 1$, $\mu^*$ decreases slower with $R$ than as predicted by (45) with $\mu^* = 0.3$ ($R = 2$) and $\mu^* = 0.075$ ($R = 10$). Outside the stability region a weak refuge ($\mu < \mu^*$) results in bounded oscillations, while a strong refuge ($\mu > 1/R$) leads to unbounded growth in population densities.

## 6.2 With host self-limitation

To prevent unbounded population growth we now include a carrying capacity. For this, we consider the previously analyzed case of the pupal parasitoid and modify model (32) to include

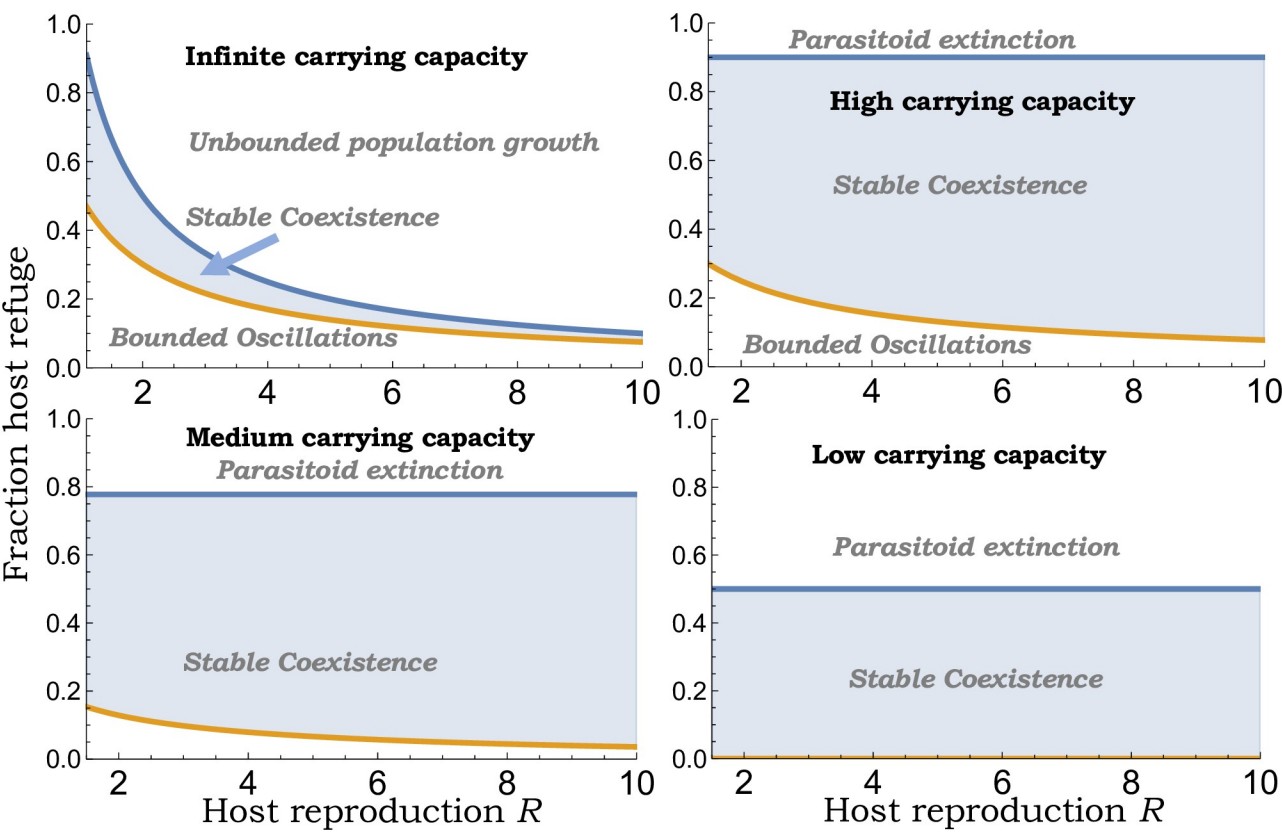

**Fig 4. The impact of host refuge on host-parasitoid population dynamics.** In the absence of any host growth limitation, stable host-parasitoid coexistence occurs for a range of refuge fractions as given by (42). Outside this range, a weak refuge results in bounded oscillations in the population densities, and a strong refuge leads to unbounded population growth. *Top-right to bottom-left to bottom-right*: In the presence of host growth limitation, the range of refuge fractions (49) allowing stable coexistence is shown with decreasing carrying capacity $K$. Here a strong refuge causes parasitoid extinction and this region expands as $K$ is lowered. The parameter space corresponding to bounded oscillation shrinks and vanishes for low-carrying capacities. For this plot, other parameters are fixed as $c_p = k = 1$ and $K = 10, 4.5, 2$.

a fixed host refuge fraction $\mu$

$$H_{t+1} = \frac{RH_t(\mu + (1 - \mu) \exp(-c_p P_t))}{1 + c_h RH_t} \tag{46a}$$

$$P_{t+1} = \frac{k(1 - \mu)RH_t(1 - \exp(-c_p P_t))}{1 + c_h RH_t}. \tag{46b}$$

Not surprisingly, with some hosts protected from wasps a higher attack rate

$$c_p > \frac{1}{kK(1 - \mu)} \tag{47}$$

is now needed for parasitoid establishment as compared with (33). With parasitoid

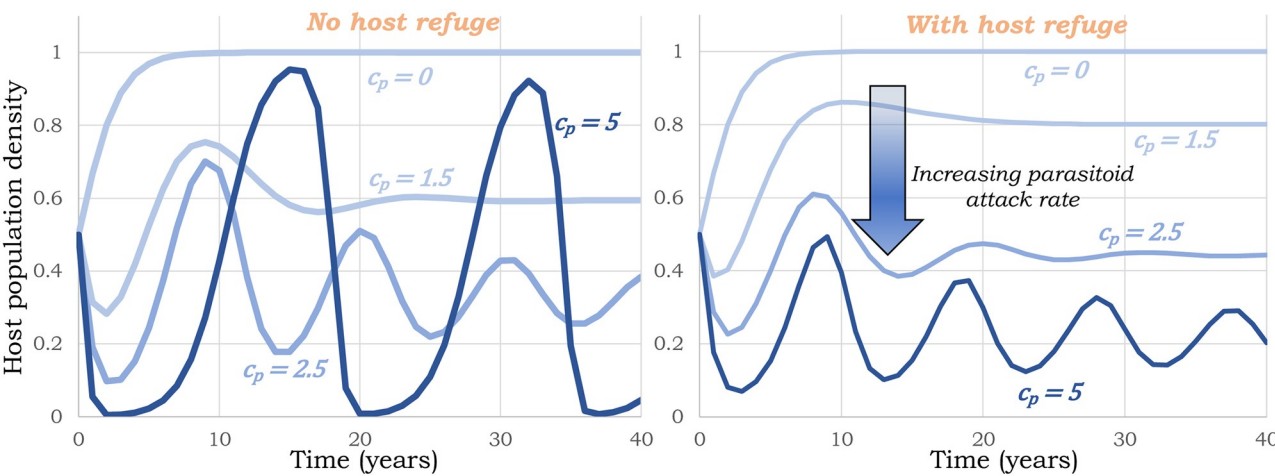

**Fig 5. Increasing attack rate destabilizes host-parasitoid population dynamics.** Trajectories of host population densities with increasing parasitoid attack rates $c_p$ (represented by darker shades of blue) as obtained from model (32) (without host refuge; left) and (46) (with host refuge; right). While in the absence of any refuge, high attack rates destabilize the population dynamics (left), with host refuge the population dynamics remain stable providing stably low suppression of the host population. Other parameters are taken as $K = k = 1$, $R = 2$, $\mu = 0$ (left) and $\mu = 0.2$ (right).

establishment, there exists a unique equilibrium

$$P^* = \frac{Kk(1 - \exp(-c_p P^*))(1 - \mu)(R(1 - \mu) - \exp(c_p P^*)(1 - R\mu))}{(R - 1)(1 + (\exp(c_p P^*) - 1)\mu)} \tag{48a}$$

$$H^* = \frac{K \exp(-c_p P^*)(R - \exp(c_p P^*) + R\mu(\exp(c_p P^*) - 1))}{R - 1} \tag{48b}$$

that is stable for a range of host refuge

$$\bar{\mu} < \mu < 1 - \frac{1}{kKc_p}. \tag{49}$$

The lower limit of host refuge need for stability satisfies $\bar{\mu} < \mu^*$, and $\bar{\mu} \approx \mu^*$ for sufficiently large $K$. These results imply that $\mu > \mu^*$ and a large enough attack rate (47) ensuring parasitoid establishment is sufficient for stable host-parasitoid population dynamics.

The stability region (49) is illustrated in Fig 4 with varying carrying capacity. Since strong limitation in the host's growth can by itself stabilize the host-parasitoid interaction in the absence of refuge, the lower limit $\bar{\mu} \to 0$ with decreasing $K$ (Fig 4; bottom right). These results have important consequences for biological control, where in the absence of host refuge a high attack rate destabilizes the coexistence equilibrium (Fig 5; left), but in the presence of host refuge $\mu > \mu^*$, the system remains stable for all values of $c_p$ with high parasitism levels driving the host population to arbitrarily low levels (Fig 5; right).

## 7 Discussion

We have analyzed a class of models where host larval density decreases throughout the stage as a result of density-dependent mortality due to intraspecific competition or parasitism. The per capita mortality rate is proportional to the host density resulting in the Beverton-Holt model describing the host population dynamics in the absence of the parasitoid [42, 56]. This is a key difference from previous work, where host population dynamics follows the Ricker model

[33]. Coupled with the Beverton-holt model is the parasitoid population dynamics leading to three different models—(18) for an egg parasitoid, (26) for a larval parasitoid, (32) for a pupal parasitoid. The larval parasitoid here refers to the scenario where density-dependent mortality acts only on the unparasitized larvae, as when mortality acts on both parasitized and unparasitized larvae, then the ecological population dynamics are identical for larval and pupal parasitoids. This latter scenario is likely when larval mortality is a result of predation by other natural enemies that do not differentiate between the two types of larvae.

We systematically analyzed these host-parasitoid models in the context of biological control of insect pests, where the primary goal is to suppress their population density via introductions of natural enemies, such as parasitoids [57–63]. Such pests can be attacked by multiple parasitoid species that could parasitize different host life-cycle stages. For example, the European corn borer *Ostrinia nubilalis* (a major pest of grains) has parasitoids that attack the egg stage (*Trichogramma ostriniae*) [64], and the larval stage (*Macrocentrus grandii* and *Lydella thompsoni*) [65]. Motivated by this we specifically contrasted population dynamic models where hosts are vulnerable to parasitoids in different developmental stages. All models (18), (26) and (32) (corresponding to parasitism of egg, larval, and pupal host stage, respectively), share a common feature of destabilized population dynamics for sufficiently large parasitoid attack rates. This can be intuitively understood from the fact that high parasitism levels drive the host density significantly below the carrying capacity, reducing (18), (26) and (32) to the unstable Nicholson-Bailey model. We quantify the limit of host density suppression across these models using $H^*/K$—the ratio of host equilibrium density (just before stability is lost for high parasitoid attack rates) and the host's parasitoid-free carrying capacity. Before summarizing this limit we discuss the minimal levels of parasitism needed for parasitoid establishment.

To assess the potential for parasitoid establishment, one should consider the product of the four dimensionless parameters $c_p kKT$. This product combines the following terms: the parasitoid attack rate (per unit time per host per parasitoid), $T$ (duration of host vulnerable stage that we assumed to be one time units earlier), $k$ (number of parasitoids per parasitized host), $K$ (host carrying capacity). The later the parasitism occurs in the host's life cycle, the higher this parameter needs to be for parasitoids to grow from small densities and establish. More specifically, parasitoid establishment requires

$$\frac{1}{R} < c_p kKT \quad \text{for egg parasitoid} \tag{50a}$$

$$\frac{R-1}{R \log R} < c_p kKT \quad \text{for larval parasitoid} \tag{50b}$$

$$1 < c_p kKT \quad \text{for pupal parasitoid.} \tag{50c}$$

Since $k = K = T = 1$ in Figs 1 & 2, the y-axis on the stability region plots can be interpreted as this dimensionless parameter $c_p kKT$. An interesting finding from our analysis is that when $R \approx 1$, the range of $c_p kKT$ allowing stable host-parasitoid coexistence is identical in all three models and given by

$$1 < c_p kKT < 3. \tag{51}$$

Here the lower limit corresponds to parasitoid establishment and is obtained by taking $R \to 1$ in (50). Crossing the upper limit $c_p kKT > 3$ destabilizes the coexistence resulting in limit cycles (Figs 1 & 2). In terms of host suppression, for $R \approx 1$ the lowest possible value of $H^*/K = 1/3$ occurs when $c_p kKT = 3$ (Fig 3). Thus, for slow-growing host populations, the timing of

parasitism may not have an appreciable impact on the population dynamics with a limit of host suppression that is 33% of the carrying capacity.

With increasing $R$, we see contrasting differences in the range of stabilizing values of $c_p kKT$, with this range contracting for an egg and larval parasitoids, but expanding for a pupal parasitoid (Figs 1 and 2). These differences directly impact the host density just before stability is again lost for high attack rates with $H^*/K \approx 0.55$ (egg parasitoid), $H^*/K \approx 0.3$ (larval parasitoid), and $H^*/K \approx 0.1$ (pupal parasitoid) for $R = 10$ (Fig 3). It is important to point out that for a fixed attack rate, the host density is the lowest for an egg parasitoid. However, for a pupal parasitoid, the coexistence equilibrium remains stable for a much broader range and larger values of attack rates leading to a lower stable host density as compared to an egg parasitoid.

How does the lower limit of $H^*/K$ depend on the form of density-dependent self-limitation in host growth? To see this, we consider a different model for host population dynamics

$$H_{t+1} = \frac{RH_t}{1 + c_h (RH_t)^q}, \quad c_h = \frac{R - 1}{(RK)^q} \tag{52}$$

where $q = 1$ is the Beverton-Holt model and we consider $q = 2$. For values of $q \in \{1, 2\}$, (52) has a stable equilibrium $H^* = K$ for all $R > 1$. When $q = 1$, the lowest stable suppression of host density varies from $\approx 20\%$ ($R = 2$) to $\approx 10\%$ ($R = 10$) in Fig 3. Our simulation results show that these limits increase to $\approx 40\%$ ($R = 2$) to $\approx 20\%$ ($R = 10$) for $q = 2$ suggesting quantitative differences in host suppression capabilities depending on the form of host growth limitation, even though the parasitoid attack rates needed for establishment are the same for $q = 1$ & 2.

We next expanded these results to consider a fraction of host refuge $0 \leq \mu < 1$. Consistent with previous analysis [13], we find that in the absence of host growth limitation, stable coexistence arises in a small range of refuge fractions (Fig 4; top-left). Our contribution here is to show that this range is approximated by

$$\frac{1}{2R} \approx \mu^* < \mu < \frac{1}{R}, \quad R \approx 1, \tag{53}$$

implying close to 50% protection for slow-growing hosts is needed for stability. The range of stabilizing $\mu$ shrinks with increasing $R$ and is given by $0.075 < \mu < 0.1$ for $R = 10$ (Fig 4). Our results show that with the inclusion of host growth limitation, this range as given by (49) dramatically increases for medium/high carrying capacities (Fig 4) and expands with increasing $R$. The dimensionless parameters $c_p kKT$ needed for parasitoid establishment is also higher by a factor of $1/(1 - \mu)$ with respect to (50). Since host refuge can stabilize the population dynamics even in the absence of any host growth limitation, above the critical refuge $\mu^* < \mu$ the system remains stable even for high attack rates that drive the host to arbitrary low densities (Fig 5). Hence, the lower limit of $H^*/K$ decreases with increasing $\mu$ and reaches zero for $\mu > \mu^*$.

In summary, our investigation reveals key insights into how the relative sequence of parasitoid attack and density-dependent host growth limitation impact the overall population dynamics. For the ecologically relevant scenario of slow-growing hosts, the system dynamics become invariant to the specific timing of parasitism. Our results coupling host refuge with the Beverton-Holt model reveal large regions of parameter space allowing stable coexistence (Fig 4) consistent with field studies implicating proportional refuges in stabilizing host-parasitoid interactions [66–70]. Future work will extend this study in several directions, such as incorporating a Type II functional response implemented using the semi-discrete approach, exploring how spatial effects alter the fundamental limits of biological control [71–75], and considering multiple parasitoid species attacking different host stages. Related to the last point, recent work has provided novel conditions for the coexistence of multiple parasitoids on a single host [76], and in many cases the requirement (6) of adult equilibrium host density

increasing with *R* emerges as a universal criterion for population stability in these complex ecological communities.

Another direction of future work would be to consider Allee effects in both the host and the parasitoid as done in recent population dynamics models [77, 78]. Inclusion of an Allee effect in the host could be interesting as it could lead to two different scenarios at high parasitism levels- one where the host is driven to extinction and the other where the population dynamics is destabilized before the Allee effect comes into play leading to parasitoid extinction. Inclusion of an Allee effect in the parasitoid should result in higher attack rates needed for parasitoid establishment than as predicted by (50), but should not impact the fundamental limits of host density suppression where the parasitoid density is relatively high. Finally, it is important to point out that while here we have used the traditional discrete-time formalism of the Nicholson-Bailey model to investigate the limits of host density suppression, it will be interesting to extend these studies with the continuous-time framework of Lotka-Volterra which is more appropriate for modeling population dynamics of insects in the tropics [1, 11].

## Supporting information

**S1 File. Mathematica file.** Wolfram Mathematica code used for generating the stability regions shown in Figs 1 and 2, and the limit of host suppression in Fig 3.
(PDF)

## Author Contributions

**Conceptualization:** Abhyudai Singh.

**Formal analysis:** Abhyudai Singh.

**Methodology:** Abhyudai Singh.

**Visualization:** Abhyudai Singh.

**Writing – original draft:** Abhyudai Singh.

**Writing – review & editing:** Abhyudai Singh.

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
