## [Decision Letter · Decision Letter 0]

9 Aug 2023

PONE-D-23-22954Limits of parasitoid-driven host density suppression: Implications for biological controlPLOS ONE

Dear Dr. Singh,

Thank you for submitting your manuscript to PLOS ONE. After careful consideration, we feel that it has merit but does not fully meet PLOS ONE’s publication criteria as it currently stands. Therefore, we invite you to submit a revised version of the manuscript that addresses the points raised during the review process.

ACADEMIC EDITOR: The evaluation of the document by both parties is complete. They emphasized the study's uniqueness and its significance for the field of biocontrol. They also advocated for a more in-depth approach to the discussion's many topics, such as the interplay between theory and pratical. I highly advise author to make improvements in response to the reviewers' recommendations. Additionally, reviewer #1 drew attention to the model's lack of detail, particularly about functional response types and other mathematical terms.

I really believe that comments will raise the quality of your article for a wider audience, and I hope to see your manuscript submitted soon.

We look forward to receiving your revised manuscript.

Kind regards,

Lucas D. B. Faria

Academic Editor

PLOS ONE

Journal Requirements:

Additional Editor Comments:

The evaluation of the document by both parties is complete. They emphasized the study's uniqueness and its significance for the field of biocontrol. They also advocated for a more in-depth approach to the discussion's many topics. I highly advise author to make improvements in response to the reviewers' recommendations. Additionally, reviewer #1 drew attention to the model's lack of detail, particularly about functional response types and other mathematical terms.

I really believe that comments will raise the quality of your article for a wider audience, and I hope to see your manuscript submitted soon.

Reviewers' comments:

Reviewer's Responses to Questions

**Comments to the Author**

1. Is the manuscript technically sound, and do the data support the conclusions?

Reviewer #1: Yes

Reviewer #2: Yes

2. Has the statistical analysis been performed appropriately and rigorously? 

Reviewer #1: N/A

Reviewer #2: N/A

3. Have the authors made all data underlying the findings in their manuscript fully available?

Reviewer #1: Yes

Reviewer #2: No

4. Is the manuscript presented in an intelligible fashion and written in standard English?

Reviewer #1: Yes

Reviewer #2: Yes

5. Review Comments to the Author

Reviewer #1: The study aimed to model temporal dynamics considering trophic interactions, focusing on the suppression limits of hosts (pests) by parasitoids. In the presented formulation, self-limitation in host growth served as a resource for population regulation in the absence of parasitoids, characterizing a density-dependent system. Different scenarios were explored for parasitism occurring at different times: before, during, and after the host species growth limitation. The models also included a fraction of hosts protected from parasitism, establishing a refuge for hosts in the system.

The results indicated that hosts with rapid growth, with parasitism occurring after host limited growth, exhibit the lowest pest density, conditioned by stable host-parasitoid coexistence. Regarding host refuge, the results suggested that specific reproduction rates lead to a critical value of protected hosts. The system's dynamics also remain stable for high parasitism rates, maintaining the host population at low sizes.

The subject is crucial for theoretical ecology and provides a foundation for implementing biological pest control programs. The theoretical basis for the model is consistent and guides the reader in understanding the system's algebraic structure. I believe the study's results have merit to be published in PlosOne. However, I would like to offer some suggestions to enhance text clarity for readers.

Firstly, consider providing background information in the introduction to familiarize readers with the subject, particularly regarding suppression limits of hosts. Given the study's objective is to investigate host density suppression limits with parasitoids, it is important to introduce basic information about the parasitoid-host relationship in the context of pests, taking into account the abilities of hosts to escape from parasitoids and the challenges of parasitoids to reach hosts in an optimal way. Maybe a rapid paragraph explaining the possible consequences of too low or high parasitism rates for the equilibrium between species, mainly when parasitoids are not generalists, and need therefore moderately act on hosts to avoid subsequently its own exclusion from system. The current introduction starts by discussing the synchrony of life stages and introduces the mathematical model without explaining the study's relevance. Also, discussing the "host refuge" concept in the introduction could create opportune expectations for readers about its importance.

While the authors mentioned plan to incorporate the Type II response in future studies, it is unclear why this issue was not addressed as clearly as the Type III response in the current manuscript. Also, regarding the first class of mechanisms, it is unclear why the parasitism escape function depended solely on parasitoid density, and not also the host density as explained in the second class of mechanisms.

The idea of a host refuge is intriguing, and the potential application scenarios mentioned are valuable. However, it might be worth considering whether the example involving parasitoid inaccessibility due to spatial location introduces the need to account for spatiality in the system. Although this point is touched upon at the end of the discussion concerning future implementations, addressing spatial location seems to leave a gap and could benefit from a referenced discussion showing real-world examples. Including more details about the examples, especially for non-technically familiar readers, could be beneficial.

Regarding the example of the host's potential immune response to parasitism, supporting it with existing literature examples is equally relevant. Referencing literature examples would also bolster the bases for inclusion of the refuge in the system since the results (Fig. 5) indicate significant dynamics differences for each scenario, affecting system stability. The caption of Figure 5 could provide more detail, identifying the curve colors and parameter 'c,' which presumably determines the parasitism rate (indicated by the downward arrow).

Lastly, approaching briefly in the article's discussion how the model could be adapted for scenarios beyond temperate regions, encompassing tropical areas with extensive plantations and high agricultural production would be valuable. This adaptation could offer critical insights for readers interested in applying the model to other scenarios, broadening the study's relevance for a wide-ranging readership.

Reviewer #2: The present work studies host-parasitoid interactions with a focus on biocontrol. The manuscript is well-written, with clear objectives, and robust results, showcasing the main points of the study. The Introduction is particularly strong, and the figures are well-presented. However, I believe the Discussion is somewhat shallow, especially in connecting the results with existing literature on biocontrol theory. While the references are extensive, their utilization could be more effective.

1. Introduction:

"A typical life cycle is illustrated in Fig. 1 where adult hosts emerge during spring and lay eggs that hatch into larvae. Host larvae feed and grow on resources over summer, and then overwinter in the pupal stage to metamorphosize as adults the following year."

-> Could you provide some examples of species that follow this specific life cycle? Certain moth species in temperate regions exhibit a different life cycle pattern from what you describe, such as the winter moth (Operophtera brumata) and the Bruce spanworm (Operophtera bruceata), in which they remain in the egg stage during winter and hatch into larvae in spring. Please specify which species or group of organisms the author is referring to, as life cycles can vary among different species.

"A multitude of mechanisms are known to stabilize population dynamics and they can be classified into two types [11, 12]".

-> The ecological significance of the two classes of mechanisms should be better explained. The first class includes the host refuge, which can be caused by some kind of protection, such as host defenses or host aggregation. The second class is caused by a Type III functional response, representing a form of population "refuge."

-> Furthermore, in the concluding remarks of your Discussion section, you reference several field studies that align with your results on the stabilizing effects of host refuges: "Our results coupling host refuge with the Beverton-Holt model reveal large regions of parameter space allowing stable coexistence (Fig. 4) consistent with field studies implicating proportional refuges in stabilizing host-parasitoid interactions [55-60]". It would be beneficial for readers if you could specify which of these studies pertain to the first class of mechanisms and which are related to the second. This distinction would provide a clearer context for understanding the ecological and biocontrol implications of your findings.

Sections 3 to 5:

I recommend the author to provide examples of pairs of species that exhibit these types of interactions. The CABI Digital Library is a valuable resource for gathering information about species with agricultural interest. Some starting examples are provided below, organized by the pest species and the corresponding parasitoids that attack different life stages:

i) Helicoverpa armigera

Egg Parasitoids: Telenomus and Trichogrammatidae (Trichogramma and Trichogrammatoidea).

Larvae Parasitoids: species of Braconidae, Ichneumonidae, and Tachinidae.

ii) Helicoverpa zea

Egg Parasitoids: Trichogramma spp.

Larvae Parasitoids: species of Braconidae and Ichneumonidae.

iii) Ostrinia nubilalis

Egg Parasitoids: Trichogramma spp.

Larvae Parasitoids: Macrocentrus grandii (Braconidae), Lydella thompsoni (Tachinidae).

iv) Manduca sexta

Larvae Parasitoids: Cotesia congregata (Braconidae).

v) Lymantria dispar

Larvae Parasitoids: Tachinid flies.

Pupal Parasitoids: Brachymeria intermedia (Chalcididae).

vi) Pieris rapae

Pupal Parasitoids: Pteromalus puparum (Pteromalidae).

vii) Plutella xylostella

Larvae Parasitoids: Diadegma insulare (Ichneumonidae), Cotesia plutellae.

viii) Halyomorpha halys

Egg Parasitoids: Trissolcus japonicus (Scelionidae), Anastatus spp.

Figs. 1--4:

Could you specify the software, methodology, and programming language employed for the two-parameter bifurcation diagrams? Ensuring that the results can be reproduced by independent researchers is essential.

7. Discussion:

A deeper exploration of the implications of biocontrol would enrich the discussion. The primary goal of any biocontrol strategy is to reduce and control pest populations, and this aspect is not well addressed in the current manuscript.

Additionally, an exploration of how the Allee effects might impact your results could enhance the manuscript. Drawing a parallel with the findings of Bompard et al. (2013) and Livadiotis et al. (2015), which you have cited (refs. [7] and [54], respectivelly), might provide valuable insights into the broader ecological context of your study.

Minor issues

The list of references is extensive. Please consider focusing on the most relevant and pertinent sources for your study.

Equation (1b) needs to close a parenthesis.

Sections 6.1 and 6.2: I recommend using the term "self-limitation" to avoid potential misinterpretations regarding external factors.

First paragraph of the Discussion section:

-> As the background of the models is the host-parasitoid system, I suggest replacing "predation" with "parasitism".

-> Please correct "Beverton-holt" to "Beverton-Holt".

Second paragraph of the Discussion section:

"The comparison between models can be intuitively understood by considering the dimensionless parameters c_p kKT that is a product for four terms".

-> I recommend rephrasing the above sentence for clarity. A more direct approach might be: "To assess the potential for parasitoid establishment, one should consider the product of the four dimensionless parameters, c_p kKT. This product combines the following terms: [...]".

In references [7] and [54], please correct the terms "alee" and "allele" to "Allee effects.

In the references, scientific names should also be italicized. See refs. [49], [56], [57].

The journals in refs. [2], [33], and [55] are in lowercase.

6. PLOS authors have the option to publish the peer review history of their article (what does this mean?). If published, this will include your full peer review and any attached files.

Reviewer #1: No

Reviewer #2: No

---

## [Author Response · Author response to Decision Letter 0]

26 Oct 2023

Please see the response to reviewers

---

## [Decision Letter · Decision Letter 1]

10 Nov 2023

PONE-D-23-22954R1Fundamental limits of parasitoid-driven host population suppression: Implications for biological controlPLOS ONE

Dear Dr. Singh,

Thank you for submitting your manuscript to PLOS ONE. After careful consideration, we feel that it has merit but does not fully meet PLOS ONE’s publication criteria as it currently stands. Therefore, we invite you to submit a revised version of the manuscript that addresses the points raised during the review process. Please submit your revised manuscript by Dec 25 2023 11:59PM. If you will need more time than this to complete your revisions, please reply to this message or contact the journal office at plosone@plos.org. Please include the following items when submitting your revised manuscript:A rebuttal letter that responds to each point raised by the academic editor and reviewer(s). You should upload this letter as a separate file labeled 'Response to Reviewers'.A marked-up copy of your manuscript that highlights changes made to the original version. You should upload this as a separate file labeled 'Revised Manuscript with Track Changes'.An unmarked version of your revised paper without tracked changes. You should upload this as a separate file labeled 'Manuscript'.If applicable, we recommend that you deposit your laboratory protocols in protocols.io to enhance the reproducibility of your results. Protocols.io assigns your protocol its own identifier (DOI) so that it can be cited independently in the future. For instructions see: https://journals.plos.org/plosone/s/submission-guidelines#loc-laboratory-protocols. Additionally, PLOS ONE offers an option for publishing peer-reviewed Lab Protocol articles, which describe protocols hosted on protocols.io. Read more information on sharing protocols at https://plos.org/protocols?utm_medium=editorial-email&utm_source=authorletters&utm_campaign=protocols.

We look forward to receiving your revised manuscript.

Kind regards,

Lucas D. B. Faria

Academic Editor

PLOS ONE

Journal Requirements:

Additional Editor Comments:

Both reviewers have finished the revision on the manuscript and pointed out that all comments were addressed successfully. However, reviewer #2 strongly suggest that all data, numerical analyses of the models and codes should be fully available for wider audience. I agree with it and kindly ask you to prepare a supplementary material or a link to it (such as https://github.com for example). I also ask to work on the most recent comments in order to improve the final version of the manuscript.

Kind regards

Reviewers' comments:

Reviewer's Responses to Questions

**Comments to the Author**

1. If the authors have adequately addressed your comments raised in a previous round of review and you feel that this manuscript is now acceptable for publication, you may indicate that here to bypass the “Comments to the Author” section, enter your conflict of interest statement in the “Confidential to Editor” section, and submit your "Accept" recommendation.

Reviewer #1: All comments have been addressed

Reviewer #2: All comments have been addressed

2. Is the manuscript technically sound, and do the data support the conclusions?

Reviewer #1: Yes

Reviewer #2: Yes

3. Has the statistical analysis been performed appropriately and rigorously? 

Reviewer #1: Yes

Reviewer #2: N/A

4. Have the authors made all data underlying the findings in their manuscript fully available?

Reviewer #1: Yes

Reviewer #2: No

5. Is the manuscript presented in an intelligible fashion and written in standard English?

Reviewer #1: Yes

Reviewer #2: Yes

6. Review Comments to the Author

Reviewer #1: I am pleased with the responses from the authors, and based on the improvements made, I believe the manuscript now provides detailed information that caters to various readers. I am in favor of accepting the manuscript for publication in PLOS ONE.

Reviewer #2: Following equation (4), the sentence "where Th is the handling time, and this has been shown to further destabilize the population dynamics" relates to the Discussion section on future research directions: "Future work will extend this study in several directions, such as incorporating a Type II functional response implemented using the semi-discrete approach". To prevent confusion, please clarify that the present study focuses on equation (3) with c_P constant. Although the Type II functional response is mentioned in the Introduction, it is not the focus of this study, which may cause misunderstandings.

Similarly, the manuscript should explicitly indicate that it investigates only the first mechanism outlined in the paragraph following equation (4). This clarification is necessary because the Type III functional response, which is associated with the second class of mechanisms, is referenced but not examined in the study.

Typos and References:

In subsection 6.1, the term "self-imitation" should be corrected to "self-limitation".

In ref. [15], please remove the full stop after "rule".

For ref. [50], please ensure "New York" is capitalized.

Please complete ref. [64] by identifying it as a "report" and adding the publisher's details.

7. PLOS authors have the option to publish the peer review history of their article (what does this mean?). If published, this will include your full peer review and any attached files.

Reviewer #1: No

Reviewer #2: No

---

## [Author Response · Author response to Decision Letter 1]

13 Nov 2023

Please see response to reviewers

---

## [Editor Report · Decision Letter 2]

4 Dec 2023

Fundamental limits of parasitoid-driven host population suppression: Implications for biological control

PONE-D-23-22954R2

Dear Dr. Singh,

We’re pleased to inform you that your manuscript has been judged scientifically suitable for publication and will be formally accepted for publication once it meets all outstanding technical requirements.

Kind regards,

Lucas D. B. Faria

Academic Editor

PLOS ONE
---

## [Editor Report · Acceptance letter]

12 Dec 2023

PONE-D-23-22954R2 

PLOS ONE

Dear Dr. Singh, 

I'm pleased to inform you that your manuscript has been deemed suitable for publication in PLOS ONE. Congratulations! Your manuscript is now being handed over to our production team.

Kind regards, 

on behalf of

Dr. Lucas D. B. Faria 

Academic Editor

PLOS ONE